# Health and Care Dependency of Older Adults in Dresden, Germany: Results from the LAB60+ Study

**DOI:** 10.3390/ijerph191811777

**Published:** 2022-09-18

**Authors:** Karla Romero Starke, Janice Hegewald, Stefanie Schmauder, Pauline Kaboth, Lena Marie Uhlmann, David Reissig, Kristin Klaudia Kaufmann, Jürgen Wegge, Gesine Marquardt, Andreas Seidler

**Affiliations:** 1Institute and Policlinic of Occupational and Social Medicine (IPAS), Faculty of Medicine, Technische Universität Dresden, 01307 Dresden, Germany; 2Federal Institute for Occupational Safety and Health (BAuA), 10317 Berlin, Germany; 3Work and Organisational Psychology, Technische Universität Dresden, 01069 Dresden, Germany; 4Social Affairs, Health and Housing Division, 10117 Berlin, Germany; 5Social and Health Care Buildings and Design, Technische Universität Dresden, 01069 Dresden, Germany

**Keywords:** healthy ageing, older adults, need for care, well-being, health-related quality of life, physical health, mental health, multimorbidity

## Abstract

As the population in Europe ages, an increased focus on the health of older adults is necessary. The purpose of the population-based LAB60+ study was to examine the current health and care situation of the population of older adults in Dresden, Germany, and to assess the effect of age, gender, and socioeconomic status (SES) on health outcomes. In the first half of 2021, 2399 out of 6004 randomly sampled residents of Dresden aged 60 years or older answered questions on their chronic conditions, care dependency, health-related quality of life (HRQoL), and well-being, among others. Of the participants, 91.6% were afflicted with at least one chronic condition, and 73.1% had multimorbidities. More than one-tenth (11.3%) of participants were care dependent. Lower levels of HRQoL and well-being were observed compared to a published German reference population, perhaps because of the ongoing COVID-19 pandemic. Gender differences were observed for some chronic health conditions, and women had a higher risk for lower HRQoL, well-being, and depressivity compared to men. A low SES was associated with a higher risk of the vast majority of health outcomes. Particularly, socioeconomic factors and gender-related inequalities should be considered for the development of prevention and health-promoting measures during late life.

## 1. Introduction

The European continent is experiencing an upward shift in its age distribution, or “population ageing”. Germany is no exception: the German Federal Statistical Office has reported that between 2020 and 2035, the number of people aged 67 (retirement age in Germany) or over will rise by 22%, from 16 million to 20 million [1]. Another indicator of the ageing of Germany’s population is the rising average age: in 2000 the average age was 40.1 years, and the expectation is that it is to rise to 49.2 years by 2050 [2]. Dresden, the capital city of the state of Saxony with a population of 561,000 inhabitants in 2021, is already experiencing a similar development. An increase in the average population age from 39.0 years in 1990 to 43.1 years in 2019 has already been documented. The share of people aged 60 to 74 years increased by about 25% during this period [3]. By 2035, the proportion of the Dresden population aged 65–74 years is expected to grow by 22%, whereas the age group 85 years and older will experience the strongest increase of 39% [4].

As the proportion of older people in the population increases, the resources and infrastructure available in cities should be better adapted to their needs and expectations. The development and growth of cities also directly impacts the lives of older persons, especially those who are socially and economically disadvantaged [5]. Therefore, the individual needs of this vulnerable population group, in particular, must also be considered in the context of urban development.

In order to reach a certain level of well-being, households with individuals with disabilities or who are care dependent face significantly higher costs compared to households without this burden [6]. This problem will be compounded in the future as the population in Europe gets older and the proportion of individuals in need of care increases. A strategy could be to prevent (or delay as much as possible) disabilities in this age group by disease prevention and by promoting healthy behaviours. Older adults also have a wide range of needs, including the desire to be able to age at home (“ageing in place”). Those who wish to do so should therefore receive sufficient support to be able to live in their own homes for as long as possible [7]. As well, a key aspect to support ageing in place is to promote health and healthy behaviours and to make the necessary resources available to do so. 

Although there are a few German-wide [8] and European-wide studies [9] dedicated to the topic of health and ageing, it is still necessary to investigate this topic at a local (city) level. Cities can directly provide resources and infrastructure to support older people to live independent, healthy, and active lives. This investment is discernable for the residents: older adults living in more age-friendly areas have reported higher life satisfaction and well-being compared to older adults living in less age-friendly areas [10,11,12]. With a city-centric study, interventions can be tailored to their residents’ needs based on the study’s specific focus and results.

There is a lack of studies on older adults in the former East Germany, a population which grew up in the German Democratic Republic (GDR) and were already adults during reunification. At the time of reunification, life expectancy in former East Germany was about three years shorter compared to that in the former West Germany [13]. Individuals in the GDR experienced some disadvantaged conditions, such as increased pollution and lower quality or shortages of health care services [13,14]. In addition, they were less able than West Germans to lead a self-determined life. It has been shown that after reunification, East Germans experienced a continued improvement in life satisfaction, partly due to greater political freedom, a change which was not seen in their West German counterparts [15]. Living conditions throughout the lifespan can have an effect on health in later life. Therefore, a study on older adults living in former East Germany is necessary.

The purpose of the study on the “Individual Living, Health and Care Situation of Older Adults in Dresden from the Age of 60” (LAB60+) was to examine the current life, health, and care situation of the population of older adults in Dresden, including assessing the effects of the COVID-19 pandemic. According to the World Health Organization (WHO) definition, health encompasses a person’s physical, as well as mental and social, well-being. Based on this comprehensive understanding of health, the LAB60+ study investigated the health status of study participants, including their physical health, chronic conditions, mental health, care situation, health-related quality of life (HRQoL), and well-being. The associations between gender, age, and socioeconomic status (SES) in the above factors is explored.

## 2. Materials and Methods

### 2.1. Study Design and Participants 

The LAB60+ study is a population-based, cross-sectional study targeting older residents (≥60 years) in the city of Dresden, Germany. People living in their own homes were included in the sample frame, whereas people living in assisted-living facilities were excluded. We aimed to have a final sample consisting of 1% of the 151,113 Dresden residents aged 60 or older (1500 people in total). For this, 6004 people were randomly selected from the official city register of residents in the first half of 2021, in the second year of the COVID-19 pandemic. Sampling was carried out following a stratified sampling strategy to ensure representativeness for the ten city districts and age groups in 5-year categories. We anticipated a 30% response for people aged 60–79 years, and the corresponding age groups (in 5-year categories) were sampled proportionally. However, we expected a lower response (20%) for the age groups 80–84 years and 85+ years, and therefore, these groups were oversampled. 

A questionnaire was sent by post to 5700 of the 6004 invited people. They had the possibility to fill out the questionnaire either in paper form or digitally. Additionally, we planned to interview a random sub-sample of 204 people (5% of the sample) in person at home. The interview consisted of the same health and need-of-care questions as the digital/paper questionnaire, but also provided an in-depth interview on other subjects, such as the participants’ living conditions, accessibility, and mobility. However, due to pandemic restrictions, they could choose between a phone interview or the digital/paper questionnaire.

The LAB60+ study was conducted conforming to the principles embodied in the Declaration of Helsinki and was approved by the Ethics Committee of the Medical Faculty of the Technical University Dresden (BO-EK-582122020). Every participant provided his or her written consent.

### 2.2. Measures

#### 2.2.1. Physical Health and Chronic Illnesses

Participants were asked to record all chronic conditions afflicting them from a list of 18. The list of conditions is shown in Appendix A. In addition, participants provided their weight and height, with which BMI values were calculated. Participants with a BMI value equal to or greater than 30 were categorized as having obesity. Chronic conditions were evaluated both separately and grouped together in categories, as shown in Table 1. Participants who reported having at least two conditions from Table 1 were classified as having multimorbidity [16,17].

#### 2.2.2. Care Dependency

Study participants reported whether they needed care and their legally recognized degree of care (1 to 5). In Germany, five care degrees make it possible to record the type and severity of impairments to independence or abilities, regardless of whether these are physical, mental, or psychological. The degree of care is determined by the long-term care insurance with the help of an assessment instrument based on nursing expertise. The care needs assessment is conducted only upon request by an individual or their relatives. The degrees of care, and thus the scope of care insurance benefits, are based on the severity of the impairments to the independence or abilities of the person in need of care. The five degrees of care are classified from minor impairments of independence or abilities (care grade 1) to the most severe impairments of independence or abilities, which are accompanied by special requirements for nursing care (care grade 5) [18]. More details on the methods used by assessors on the classification of care degrees in Germany is shown in the Appendix A.

#### 2.2.3. Health-Related Quality of Life (HRQoL), Well-Being, and Depressivity

The Short-Form Health Survey-8 (SF-8), a short form of the SF-36, was used for measuring health-related quality of life (HRQol) [19,20]. The SF-8 measures eight dimensions of HRQoL: physical functioning (PF), role-physical (RP), bodily pain (BP), general health (GH), vitality (VT), social functioning (SF), role-emotional (RE) and mental health (MH). Two second-order factors were calculated using the methods indicated in the SF-8 Handbook: the physical component score (PCS) and the mental component score (MCS) [20]. Higher values indicate better physical/mental health, and interpretation relies on comparison with norm-based scores [19,21]. We compared the LAB60+ SF-8 scores with a norm sample of the German general population [19].

For the evaluation of well-being, the 5-item WHO Well-Being Index (WHO-5) was used [22]. The five questions relate to the past two weeks and are evaluated with a 6-point scale. All five questions must be answered to build the index, which has a range from 0 to 25 points. Higher values indicate better subjective well-being. The raw values were multiplied by four in order to facilitate comparisons between populations/studies [22] (range 0–100). Furthermore, the WHO-5 is used as a screening instrument to detect depression [23]. A raw score of below 13 (out of a maximum of 25) is seen as an indicator for the possible presence of major depression [23]. However, since the WHO-5 cannot be used to make a medical diagnosis of depression, individuals who score below 13 are referred to as having increased depressive symptoms, to which we will refer to as “depressivity” in this report.

#### 2.2.4. Sociodemographic Factors

The LAB60+ participants answered questions on their age (year of birth), gender according to their identity card (male, female, or other), family status, education, monthly net income (in categories), and country of birth. SES was derived by the Winkler Index [24] using income and education data. 

### 2.3. Statistical Analysis

Weighting for the age groups was carried out through post-stratification weights [25] to reflect the age structure of the Dresden population aged 60 years and older. This was necessary because the response for the younger age groups was less than expected, whereas the response for the older age groups was more than expected.

Descriptive analyses were stratified by age, gender, and SES. Because the outcomes of interest were common for this population and odds ratios overestimate relative risks, for binary outcomes, a Poisson regression was used instead of logistic regression [26]. Multivariate linear regression was used to analyse continuous outcomes (PCS-8, MCS-8, and WHO-5). All regressions were adjusted for age, gender, and SES.

## 3. Results

### 3.1. Participants’ Response and Characteristics

In total, 2399 people responded to the survey (40% response), either by paper (95%), online (4%), or through a telephone interview (1%). Response was lower for the age groups 60–64 years (35.6%), 80–84 years (39.9%), and 85+ years (33.6%) compared to the other age groups, which had similar responses (42.5%, 45.6%, and 41.5% for age groups 60–64, 70–74, and 75–79 years, respectively). Responses for men and women were similar for all the age groups, except for the highest two age groups (80–84 years: men 44.8%, women 36.1%; 85+ years: men 46.2%, women 25.8%).

Table 2 shows the weighted characteristics of the LAB60+ participants by age group and gender. Of the participants, 52.5% were women. There were not enough people (nine total) reporting a diverse gender to report stratified results of this particular category. The average age was 74.1 years (range 60 to 100 years). The majority of participants were born in Germany (94.9%). Regarding SES, 29.2% of participants had a high SES, 45.4% had a middle SES, and 15.5% had the lowest SES. A higher proportion of women belonged to the lowest SES (19.8% vs. 11.1%), and SES decreased with increasing age.

### 3.2. Chronic Conditions

The majority of participants (90.5%) reported suffering from at least one chronic condition (Table 3). The most frequently cited category comprised cardiometabolic conditions at a prevalence of 62.9% (with hypertension being the most prevalent out of this group of diseases at 52.3%). The next most prevalent chronic conditions were musculoskeletal diseases (51.6%) and sensory impairments (35.0%). The most common reported musculoskeletal conditions were osteoarthritis (31.2%) and chronic pain (30.8%). Further, eye disease (e.g., cataracts) and hearing impairment were similarly prevalent in the sensory impairment category (22.8% and 20.6%, respectively). Compared to women, men had a higher prevalence of cardiovascular disease (CVD) in general (24.2% vs. 14.9%). In particular, for CVD, the prevalence of coronary heart disease (13.9% vs. 7.5%) and cardiac infarction (6.2% vs. 2.2%) was higher for men than for women. Moreover, men had a higher prevalence of cancer (11.5% vs. 6.8%) and hearing impairment (23.6% vs. 17.9%) compared to women. On the other hand, compared to men, women were more often impacted by depression (8.8% vs. 4.8%), chronic pain (37.2% vs. 23.6%), and osteoporosis (19.2% vs. 5.1%).

Figure 1 depicts the disease categories by age groups separately for women and men. In both genders, an increase across age groups was observed for cardiometabolic conditions, cardiovascular disease, musculoskeletal disease, sensory impairment, and incontinence, although the upward trend in men for cardiometabolic conditions was less steady. 

After adjusting for age and SES, men were more likely than women to have cardiometabolic conditions (diabetes and hypertension), cardiovascular disease (coronary heart disease, cardiac infarction, and stroke), cancer, and hearing impairment (Table 4). On the other hand, women were more likely than men to have all conditions under the musculoskeletal disease category, depression, incontinence, and dementia. Except for diabetes, obesity, lung disease, depression, and rheumatism, there was a strong age gradient in the relative risks after adjustment for gender and SES. Furthermore, participants having a low SES had a higher prevalence of chronic conditions compared to participants with a high SES, except for eye disease, hearing impairment, and osteoporosis. Statistical significance in the association between SES and chronic disease was reached for all cardiometabolic conditions, some musculoskeletal conditions (arthrosis and chronic pain), and dizziness. 

On average, participants had three chronic conditions, and 73.1% the participants had two or more chronic conditions. After adjusting for age and SES, men were slightly more at risk for having multimorbidities compared to women (RR = 1.05, 95% CI 0.98–1.11), but the association was not statistically significant (Table 5). The risk of having multimorbidities increased with increasing age and was highest for people with middle and low SES (RR_middleSES_ = 1.10, 95% CI 1.03–1.17; RR_lowSES_ = 1.15, 95% CI 1.07–1.24).

### 3.3. Care Dependency

Slightly more than one-tenth (11.3%) of participants reported having some sort of care dependency (Table 6). The proportion of women with a care dependency was higher compared to men (13.7% vs. 8.7%). As well, care dependency increased with increasing age and decreased SES. Almost half of participants (47.2%) had a degree of care of 2 (significant impairment of independence or abilities), followed by a degree of 3 (severe impairment of independence or abilities, 27.1%), 1 (minor impairments of independence or abilities, 21.8%), and 4 (most severe impairment of independence or abilities, 3.4%). Only 0.3% of participants had a degree of care of 5, or the most severe impairments of independence or abilities with special requirements for nursing care. 

After adjusting for age and SES, men were only very slightly less likely to be care dependent compared to women (RR = 0.97; 95% CI 0.95–0.99) (Table 4). Care dependency increased with age. Participants having a middle or low SES were more likely to be care dependent than participants with a high SES (RR_middleSES_ = 1.04, 95% CI 1.02–1.06; RR_lowSES_ = 1.12, 95% CI 1.08–1.16).

### 3.4. Health-Related Quality of Life and Well-Being

Mean PCS-8 (physical health dimension) and MCS-8 (mental health dimension) scores were 44.5 and 48.2, respectively (Table 7). Subscales for the SF-8 are also shown in Table 4, and the results stratified by age and gender are presented in Appendix A. Appendix A shows the results of the subscales compared to the German population’s normal sample [19]. Compared to the normal sample, health-related quality of life was, in general, slightly lower for this population. Although the physical health dimension for the lowest age groups (60–79 years) was comparable to the normal sample, for the highest age groups (70–79 years and 80+ years), it was slightly lower than the normal sample (70–79 years—women: 43.9 vs. 44.8 points, men: 44.9 vs. 47.2 points; 80+ years—women: 37.9 vs. 41.7 points, men: 41.4 vs. 44.5 points). The mean PCS-8 scores decreased with increasing age and were higher for men than for women. 

The mental health dimension was lower for our population compared to the normal sample across all age groups and genders (see Appendix A). The difference in the mental health dimension between both populations was between 10% and 15%. Like the PCS-8, the MCS-8 scores were higher for men than women. In terms of age, a U-curve relationship was observed: starting at the age group of 65–69 years, the MCS-8 scores increased and then decreased steadily starting from the age group of 75–79 years.

After adjusting for age and SES, there was no statistically significant difference between men and women regarding PCS-8 scores (β = 0.36; 95% CI = −0.51, 1.22). Physical health-related quality of life decreased with increasing age, and those in the middle and low SES categories were more likely than those in the high SES category to have lower physical health-related quality of life (β_middleSES_ = −2.59, 95% CI: −3.53, −1.64; β_lowSES_ = −5.11, 95% CI: −6.44, −3.78) (Table 8). A similar effect of SES was observed for mental health-related quality of life: persons in the middle and low SES categories had lower quality of life compared to persons in the high SES category: (β_middleSES_ = −2.00, 95% CI: −3.00, −0.99; β_lowSES_ = −3.17, 95% CI: −4.59, −1.75). Men had a higher mental-health quality of life compared to women (β = 2.43, 95% CI: 1.50, 3.36). The age groups of 65–74 years and 75–79 years had a higher mental-health quality of life compared to the lowest age group (60–64 years). 

The mean WHO-5 score was 58.3. Appendix A presents the results stratified by gender and age. Men were more likely to have a higher well-being score compared to women (β = 5.16, 95% CI: 3.06–7.26) (Table 8). Participants in the age group of 75–79 years were more likely to experience higher well-being scores than persons in the reference age group of 60–64 years (β_75–79 years_ = 3.63, 95% CI: 0.13, 7.14). On the other hand, people in the highest age groups (80–84 years and 85+ years) were more likely to have a worse well-being than participants in the reference age group (β_80–84 years_ = −3.74, 95% CI: −7.31, −0.17; β_85+ years_ = −5.72, 95% CI: −9.86, −1.59). 

One-third (33.7%) of participants had depressivity, or a possible depression (Table 7). Women were more likely than men to have depressivity (38.1% vs. 28.9%), but after adjusting for age and SES, men had a lower risk than women (OR = 0.78; 95% CI: 0.68–0.89). Like the MCS-8, the WHO-5-based risk of a possible depression showed a U-curve relationship with respect to age: the age groups of 65–69 years and 70–74 years had the lowest risk, whereas the highest age groups (75 years and over) had increasingly higher risks, although only the age group 85+ reached statistical significance (OR_85+years_ = 1.28; 95% CI: 1.01–1.63). Further, the risk of depressivity increased with decreasing SES (Table 7).

## 4. Discussion

This population-based study allowed for an overview of the health status of Dresden residents aged 60 years or older living in private domiciles. In summary, we found that most of the participants were afflicted with at least one chronic condition, and close to three-quarters had multimorbidities, defined as having two or more chronic conditions. Further, one-tenth of participants were in need of care. Men had higher levels of mental health and higher levels of well-being compared to women, after adjustment for SES and age. We observed an inverse U-trend for the association between age and well-being: the age group of 65–74 enjoyed the highest levels of well-being, whereas the higher age groups (80+ years) had the lowest levels. After adjusting for age and gender, the lowest SES group had the highest presence of chronic diseases and care dependency, and the lowest levels of well-being and health-related quality of life.

### 4.1. Chronic Conditions and Multimorbidity

Comparisons on chronic conditions and health assessments are difficult due to different study periods and study populations, including ages studied, assessment of conditions, and recruitment procedures. To provide context for our results, we tried to use representative German surveys such as the German Health Update (GEDA) [27] and the German Ageing Survey (DEAS) [28].

Regarding cardiometabolic conditions, this study observed point-prevalences increasing by age of 50–70% for women and 60–75% for men. For women, these numbers are lower than the GEDA 2009 telephone survey, which reported a prevalence of cardiometabolic conditions of 75% in women ages 65–74 years and 80% in women aged 75+ years (75% to 80%) [29]. The same survey reported a prevalence of cardiometabolic conditions of around 70% in men in the age group 65+, in accordance with our study. The GEDA survey included hypercholesterolemia in their definition of cardiovascular conditions, which may partly explain lower prevalences in women. A further reason for this discrepancy is that the GEDA survey, unlike our study, included taking medications to lower blood pressure in the definition of hypertension [29,30]. The prevalence of hypertension was increased by 2.5% when taking into account self-reported blood pressure medication [30]. The prevalence of obesity in our study (20%) was in agreement with the DEAS survey (21.2%) [31]. Likewise, the prevalence of diabetes for our study (10–23%, increasing in age) was similar to that reported in the GEDA 2009 study (11–19% for similar age groups) [32].

Similar to this study, the GEDA survey reported a lower prevalence of CVD for women compared to men. In this study, for women, prevalences varied between 6% to 31%, depending on the age group, which was in agreement with the GEDA results. Furthermore, for this study, the presence of CVD varied from 13% to 44% for men, which was also in agreement with the GEDA survey.

The prevalence of cancer in this study was lower than that of the GEDA study: for our study, prevalences for women varied between 6% and 12%, depending on the age, whereas men had prevalences of 7% to 16%. The GEDA study reported prevalences of close to 20% for both genders. The difference in results is most likely due to our definition, as we asked about cancer in the past 5 years, whereas the GEDA survey inquired on the lifetime prevalence of cancer.

Only slightly higher prevalences of musculoskeletal disease were observed for this study (47% to 81% for women; 41% to 52% for men) compared to the GEDA study (women 65+ years: 63%; men 65 to 75+ years: 40 to 45%). In terms of lung disease, sensory impairments, and depression, the results of this study tended to mirror the GEDA results.

Almost three-fourths (73.1%) of respondents had multimorbidities, which is slightly higher than the DEAS [33] (68% multimorbidity) and GEDA results [34] for similar age groups. Like Puth and colleagues, our study found an increased risk of multimorbidity with increased age and decreased SES [34]. There is difficulty comparing multimorbidity across different studies due to the lack of consensus on the definition and measurement of comorbidity [35,36,37]. A review found that the most oft-used definition used a cut-point of two or more conditions (diseases, risk factors, or symptoms), which is analogous to how we defined comorbidity [37]. Nonetheless, the definition of multimorbidity used ranges from different cut-off points used (either not specified or ≥2 conditions or ≥3 conditions) to include either chronic diseases only or include risk factors and symptoms as well [35,37]. Therefore, it is necessary to keep these discrepancies in mind when asserting prevalences of multimorbidity in the population.

### 4.2. Care Dependency

Close to one-tenth (11.3%) of respondents reported having a recognized care dependency. These values are akin to what was reported by the German Federal Statistical Office for people who were cared for at home in 2019 [38]. Regarding the distribution of persons according to the severity of their care level, care levels 2 and 3 occurred most frequently in the study population (47.1% and 27.1%, respectively) and is also comparable to the situation in Germany as a whole [38]. 

Similar to the German national statistics [39], the prevalence of women with a care dependency was higher than men. There may be several reasons for this. First, men have a shorter life expectancy than women. However, especially healthy men may live to a very old age, and these men may require less care than women of the same age [40]. Second, for our study, the proportion of women living alone after age 70 is about twice as high as that of men (see Appendix A); again, possibly due to the fact that men have a shorter life expectancy and that women tend to have older partners [40]. When care is needed, women may therefore need to apply for care more quickly [40], whereas men in need of care are often initially cared for by their partners [41]. Since the care needs of these men are covered by their wives, they usually do not apply for care benefits and are therefore included less frequently in the official care statistics [40].

The risk of being care dependent, after adjustment for age and sex, increased with decreased SES. These results are in accordance with studies finding that people in privileged socioeconomic groups have fewer years of disability despite a longer life expectancy [42,43,44,45,46].

### 4.3. Health-Related Quality of Life (HRQoL) and Well-Being

Compared to the German reference sample, the physical health dimension scores were comparable for the lowest age groups, but starting at 70 years, the scores were about 10% lower than the reference sample. The mental health dimension scores were about 10% lower for our study compared to the reference sample, although for women aged 80+ years, the difference was around 15%. Similarly, the WHO-5 well-being scores for this study were lower than those of a German reference sample in 2004 using a similar age group (≥61 years), (58. 3 vs. 66.8 points) [22]. Moreover, about one-third of the participants had depressivity. The reason for the lower health-related quality of life and well-being seen for this study may lie in the timeframe of the study, which was during the third wave of the COVID-19 pandemic in Germany. During this period, there were several government-imposed contact restrictions, such as store and school closings, and bans on gatherings and on recreational offerings. There was additional frustration expressed by the 80+ year old participants, as the COVID-19 vaccine appointments were offered mostly online, and they could not get through on the phones for an appointment. Such restrictions had a negative impact on health and well-being: when comparing pre-pandemic time points to the lockdown phases, a general increase in mental distress in the general population was observed [47]. A similar effect was also seen in a population of older German adults [48]. For this study, it is not possible to disentangle the effect of the COVID-19 pandemic on the health-related quality of life and well-being with the typical state of well-being in this population. However, the well-being scores are comparable to those of a German study conducted in a similar timeframe during the pandemic [49]. 

Nonetheless, the general associations of sex, age, and SES on HRQoL and well-being is worth discussing. Similar to our study, an inverse U-shaped relationship between well-being and age has been observed in high-income countries [50,51]. The upward trend in our study after the retirement years (60–65 years) could be due to the retirement transition. Retirement may improve health, including psychological distress and depressive symptoms, but the change depends on the context of the work conditions and an individual’s educational and socioeconomic status [52,53]. As participants age past this post-retirement stage and health deteriorates, a drop in well-being and HRQoL follows, perhaps reflecting the end of life [54]. Similar to our study, studies in older adults reported lower well-being for women than for men [55]. Further, the association between socioeconomic status on health is well-known [56], and we observed it in this population of older adults.

### 4.4. Strengths and Weaknesses

LAB60+ is the first study investigating the health of older adults in the city of Dresden. A main strength of this study is its representativeness. The city’s ten districts and the different age groups and genders were well-represented, which minimized selection bias. Where possible, we used validated instruments, such as for well-being and HRQoL. 

However, our results should be considered in light of their limitations. Some outcomes, such as chronic conditions and care dependency, were self-reported, and some information bias can have occurred. Nevertheless, comparisons to German statistical reports show an agreement in values in terms of morbidity and care dependency. Since the cross-sectional survey took place during a lockdown phase of the pandemic, it is difficult to judge whether the HRQoL and well-being results were affected by the pandemic. 

This study aimed to explore correlations and not causality, as the cross-sectional design limits any causal interpretation. An example would be the two-way causality relationship between socioeconomic status and health. 

### 4.5. Public Health Implications

The results of our study underline the importance of a broad health evaluation in older adults. The high prevalence of multimorbidity indicates a need for coordinated care as well as preventive strategies to reduce overall morbidity and increase the quality of remaining life. Since lower well-being was probably largely due to the pandemic situation, city-wide measures targeting older adults should be developed to overcome or compensate for COVID-19 restrictions, especially as those in the lowest socioeconomic classes are particularly vulnerable to adverse health outcomes, and prevention measures should be targeted to this vulnerable group. At the city level, possible interventions include age-appropriate physical activity programs and awareness campaigns regarding existing services such as the Senior Hotline and senior meeting groups. At the city level, the promotion and, if necessary, the building of walking paths, parks, or green spaces for physical activity can be useful for the improvement of physical and mental health.

## 5. Conclusions

In this first representative study on older adults in Dresden, Germany, we evaluated their physical and mental health, health-related quality of life, and well-being. Our results can be used for city-wide prevention programs focusing on targeted health promotions. In our study, most adults had at least one chronic condition, and more than two-thirds had multimorbidity. Participants rated their well-being and HRQoL lower than the German reference population, and about one-third had depressivity, which may be due to the effects of the pandemic. Future studies should target interventions to mitigate the effect of the pandemic (or future pandemics) on older adults’ mental health and well-being. 

We observed age and sex differences, as well as an increased risk for the vast majority of health outcomes with decreasing SES. Health promotion and prevention measures should focus on the reduction of socially determined and gender-related inequalities in health opportunities. Prevention programs should target or continue to target individuals belonging to a lower socioeconomic status. 

This study also provides a baseline for a longitudinal study to evaluate whether any subsequent communal interventions improve the health of the residents, and if the needs of these residents have changed. 

## Figures and Tables

**Figure 1 ijerph-19-11777-f001:**
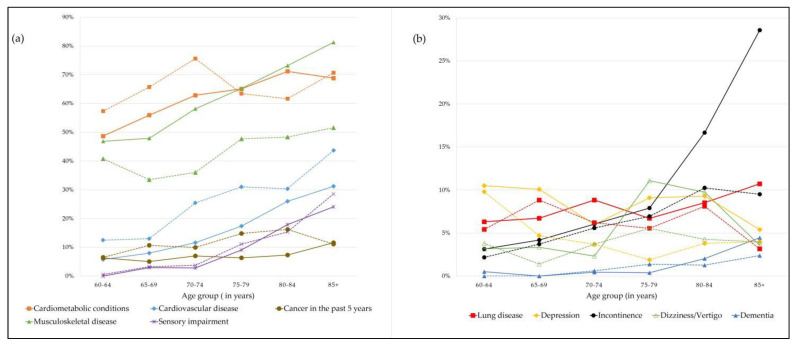
Prevalence of disease categories by age groups for women (solid lines) and men (dotted lines). (**a**) Disease categories: Cardiometabolic conditions, cardiovascular disease, cancer in the past five years, musculoskeletal disease, and sensory impairment. (**b**) Disease categories: lung disease, depression, incontinence, dizziness/vertigo, and dementia.

**Table 1 ijerph-19-11777-t001:** Individual chronic conditions and disease categories.

Chronic Condition	Disease Category
Hypertension, diabetes, obesity (BMI ≥ 30)	Cardiometabolic conditions
Myocardial infarction, chronic heart failure, stroke, coronary heart disease	Cardiovascular disease
Osteoarthrosis, rheuma, osteoporosis, chronic pain (e.g., back pain)	Musculoskeletal disease
Depression	Depression
Dementia (e.g., Alzheimer’s)	Dementia
Lung disease (e.g., COPD)	Lung disease
Eye disease (e.g., cataract), hearing impairment	Sensory limitations
Cancer (in the past 5 years)	Cancer
Incontinence	Incontinence
Dizziness/Vertigo	Dizziness

**Table 2 ijerph-19-11777-t002:** Characteristics of survey respondents, after weighting, by age group and gender.

Characteristics	All	By Age Group						By Gender	
	60+ Years	60–64 Years	65–69 Years	70–74 Years	75–79 Years	80–84 Years	85+ Yars	Women	Men
	(N = 2399)	(N = 376)	(N = 455)	(N = 379)	(N = 470)	(N = 480)	(N = 239)	(N = 1254)	(N = 1136)
	100%	19.6%	19.9%	15.0%	20.3%	15.7%	9.5%	52.5%	47.6%
	Wtd. %	Wtd. %	Wtd. %	Wtd. %	Wtd. %	Wtd. %	Wtd. %	Wtd. %	Wtd. %
**Gender**									
Women	52.5	50.8	52.5	57.2	53.9	51.3	47.1	n/a	n/a
Men	47.6	49.2	47.5	42.8	46.1	48.8	52.9	n/a	n/a
**Age Group**									
60–64 years	n/a	n/a	n/a	n/a	n/a	n/a	n/a	19.0	20.3
65–69 years	n/a	n/a	n/a	n/a	n/a	n/a	n/a	19.9	19.9
70–74 years	n/a	n/a	n/a	n/a	n/a	n/a	n/a	16.3	13.4
75–79 years	n/a	n/a	n/a	n/a	n/a	n/a	n/a	20.9	19.7
80–84 years	n/a	n/a	n/a	n/a	n/a	n/a	n/a	15.4	16.2
85+ years	n/a	n/a	n/a	n/a	n/a	n/a	n/a	8.5	10.6
**Family status**									
Single	4.7	7.8	6.0	4.6	3.2	1.9	3.8	4.3	5.1
Married	69.1	72.1	72.3	73.7	73.4	63.4	49.6	60.4	78.9
Divorced	10.7	16.1	14.0	9.1	7.6	7.5	6.7	12.7	8.2
Widowed	15.5	4.0	7.8	12.6	15.8	27.2	39.9	22.5	7.8
**Country of birth**									
Germany	94.9	96.0	97.1	97.6	91.4	93.5	93.7	95.1	94.7
Other	5.1	4.0	2.9	2.4	8.6	6.5	6.3	4.9	5.3
**SES**									
Low	15.5	10.6	8.5	13.5	15.5	23.8	33.7	19.8	11.1
Middle	45.4	43.5	49.5	54.1	50.7	35.6	29.3	49.9	40.3
High	39.2	45.9	42.1	32.4	33.8	40.6	37.0	30.3	48.6

Wtd.% = weighted %. Cell percentages are based on weighted data for everyone in that age group. N at top of each column is the unweighted number of respondents in that group. n/a = not applicable.

**Table 3 ijerph-19-11777-t003:** Self-reported chronic conditions of LAB60+ participants, after weighting, by gender.

Chronic Condition	All	Women	Men
	(N = 2219)	(N = 1166)	(N = 1053)
	Wtd. % (95% CI)	Wtd. % (95% CI)	Wtd. % (95% CI)
No chronic conditions	8.4 (7.4–9.6)	8.7 (7.2–10.4)	8.2 (6.8–10.0)
Cardiometabolic conditions	62.9 (60.9–64.8)	61.0 (58.3–63.7)	64.8 (61.9–67.5)
Diabetes	18.2 (16.7–19.8)	16.3 (14.4–18.4)	20.2 (18.0–22.7)
Hypertension	52.3 (50.4–54.3)	50.9 (48.2–53.7)	54.0 (51.5–56.9)
Obesity	20.4 (18.9–22.1)	21.6 (19.3–24.0)	19.0 (16.8–21.4)
Cardiovascular disease	19.2 (17.7–20.8)	14.9 (13.0–17.0) *	24.2 (21.8–26.7) *
Heart failure	7.5 (6.5–8.7)	6.7 (5.5–8.2)	8.5 (7.0–10.3)
Coronary heart disease	8.9 (8.8–9.1)	7.5 (6.2–9.1) *	13.9 (12.0–16.0) *
Cardiac infarction	4.1 (3.4–5.0)	2.2 (1.5–3.2) *	6.2 (5.0–7.8) *
Stroke	3.8 (3.2–4.6)	2.8 (2.0–3.9)	4.9 (3.8–6.3)
Lung disease	7.0 (6.1–8.1)	7.6 (6.3–9.2)	6.4 (5.1–8.0)
Cancer (in the past 5 years)	9.0 (7.9–10.2)	6.8 (5.5–8.3) *	11.5 (9.8–13.5) *
Musculoskeletal disease	51.6 (49.6–53.6)	59.7 (57.0–62.4)	42.4 (39.6–45.3)
Arthrosis	31.2 (29.4–33.1)	37.3 (34.7–40.0) *	24.3 (21.9–26.9) *
Rheumatism	4.8 (4.0–5.7)	5.8 (4.6–7.2)	3.8 (2.8–5.0)
Osteoporosis	12.6 (11.3–13.9)	19.2 (17.2–21.5) *	5.1 (4.0–6.6) *
Chronic pain (e.g., back pain)	30.8 (28.9–32.6)	37.2 (34.5–39.9) *	23.6 (21.1–26.1) *
Depression	7.0 (6.0–8.0)	8.8 (7.3–10.5) *	4.8 (3.7–6.2) *
Sensory impairment	35.0 (33.1–37.0)	34.6 (32.0–37.3)	35.5 (32.8–38.3)
Eye disease	22.8 (21.1–24.5)	24.5 (22.2–26.9)	20.9 (18.6–23.3)
Hearing impairment	20.6 (19.0–22.2)	17.9 (15.9–20.1) *	23.6 (21.2–26.1) *
Incontinence	7.6 (6.6–8.7)	9.1 (7.6–10.8) *	6.0 (4.7–7.5) *
Dizziness/vertigo	11.4 (10.2–12.7)	13.4 (11.6–15.4) *	9.3 (7.7–11.1) *
Dementia	0.9 (0.6–1.3)	1.0 (0.5–1.7)	0.8 (0.4–1.5)

Wtd.% = weighted %; CI = confidence intervals. Cell percentages are based on weighted data for everyone in that age group. N at top of each column is the unweighted number of respondents in that group. * Statistically significant difference (95% CIs do not overlap).

**Table 4 ijerph-19-11777-t004:** Risk of self-reported chronic conditions from Poisson regression, adjusted for gender, age, and socioeconomic status (SES).

	Gender	Age Group					SES	
Chronic Condition	Men vs. Women (Ref.)	65–69 yrs. vs. 60–64 yrs. (Ref.)	70–74 yrs. vs. 60–64 yrs. (Ref.)	75–79 yrs. vs. 60–64 yrs. (Ref.)	80–84 yrs. vs. 60–64 yrs. (Ref.)	85+ yrs vs. 60–64 yrs. (Ref.)	Middle vs.High SES (Ref.)	Low vs. High SES (Ref.)
	RR (95% CI)	RR (95% CI)	RR (95% CI)	RR (95% CI)	RR (95% CI)	RR (95% CI)	RR (95% CI)	RR (95% CI)
Cardiometabolic conditions	1.20 (1.02–1.18)	1.19 (1.04–1.36)	1.33 (1.17–1.52)	1.24 (1.09–1.42)	1.27 (1.12–1.45)	1.36 (1.18–1.57)	1.11 (1.03–1.21)	1.16 (1.04–1.28)
*Diabetes*	1.38 (1.13–1.69)	1.66 (1.09–2.53)	2.62 (1.76–3.92)	1.94 (1.29–2.92)	2.26 (1.51–3.38)	2.39 (1.53–3.72)	1.45 (1.16–1.83)	1.58 (1.18–2.11)
*Hypertension*	1.11 (1.02–1.22)	1.18 (1.00–1.39)	1.36 (1.15–1.60)	1.37 (1.17–1.60)	1.42 (1.21–1.66)	1.47 (1.23–1.75)	1.14 (1.04–1.26)	1.09 (0.95–1.23)
*Obesity*	0.97 (0.80–1.17)	0.88 (0.67–1.16)	1.03 (0.78–1.36)	0.69 (0.51–0.93)	0.50 (0.35–0.70)	0.53 (0.35–0.80)	1.17 (0.94–1.45)	1.77 (1.37–2.29)
Cardiovascular disease	1.70 (1.40–2.07)	1.12 (0.71–1.77)	1.96 (1.28–3.00)	2.64 (1.78–3.90)	2.98 (2.02–4.39)	2.98 (2.02–4.39)	1.06 (0.86–1.31)	1.23 (0.95–1.59)
*Heart failure*	1.25 (0.89–1.76)	1.70 (0.77–3.74)	2.00 (0.89–4.46)	3.28 (1.60–6.76)	4.24 (2.10–10.93)	5.26 (2.53–10.92)	1.08 (0.75–1.55)	1.11 (0.70–1.77)
*Coronary heart disease*	2.01 (1.50–2.67)	1.38 (0.70–2.71)	2.32 (1.22–4.42)	2.92 (1.60–5.32)	3.59 (1.99–6.46)	4.64 (2.53–8.48)	1.30 (0.95–1.78)	1.62 (1.11–2.34)
*Cardiac infarction*	2.90 (1.75–4.79)	0.58 (0.21–1.60)	1.98 (0.88–4.46)	1.96 (0.91–4.24)	1.92 (0.88–4.15)	2.20 (0.94–5.12)	0.88 (0.54–1.43)	1.03 (0.55–1.94)
*Stroke*	1.72 (1.05–2.81)	0.65 (0.23–1.84)	1.72 (0.72–4.14)	1.35 (0.56–3.25)	1.90 (0.82–4.43)	3.38 (1.44–7.93)	0.91 (0.55–1.53)	1.43 (0.76–2.71)
Lung disease	0.82 (0.58–1.18)	1.32 (0.77–2.27)	1.19 (0.67–2.13)	0.78 (0.42–1.46)	1.29 (0.74–2.26)	0.99 (0.48–2.05)	1.11 (0.75–1.65)	1.56 (0.94–2.57)
Cancer (in the past 5 years)	1.66 (1.23–2.23)	1.04 (0.61–1.79)	1.34 (0.78–2.31)	1.49 (0.91–2.46)	1.77 (1.09–2.88)	1.75 (1.00–3.06)	0.98 (0.72–1.34)	1.19 (0.79–1.79)
Musculoskeletal disease	0.72 (0.65–0.79)	0.94 (0.79–1.11)	1.02 (0.86–1.21)	1.27 (1.09–1.63)	1.40 (1.21–1.63)	1.49 (1.27–1.75)	1.17 (1.06–1.30)	1.20 (1.06–1.36)
*Arthrosis*	0.65 (0.56–0.75)	0.94 (0.73–1.19)	1.03 (0.80–1.32)	1.19 (0.94–1.49)	1.52 (1.21–1.90)	1.56 (1.21–2.01)	1.08 (0.88–1.32)	1.20 (1.03–1.39)
*Rheumatism*	0.83 (0.54–1.28)	0.71 (0.31–1.62)	0.45 (0.16–1.25)	2.21 (1.15–4.29)	1.74 (0.87–3.47)	1.07 (0.45–2.58)	1.23 (0.75–2.01)	1.77 (0.99–3.15)
*Osteoporosis*	0.25 (0.18–0.35)	1.26 (0.71–2.21)	1.47 (0.83–2.61)	2.35 (1.42–3.90)	2.51 (1.50–4.20)	4.46 (2.67–7.46)	1.18 (0.62–1.14)	1.18 (0.84–1.66)
*Chronic pain* *(e.g., chronic back pain)*	0.66 (0.57–0.77)	0.87 (0.67–1.13)	0.84 (0.64–1.11)	1.37 (1.09–1.72)	1.39 (1.10–1.76	1.57 (1.21–2.02)	1.32 (1.12–1.56)	1.43 (1.17–1.76)
Depression	0.51 (0.35–75)	0.86 (0.54–1.36)	0.45 (0.25–0.83)	0.47 (0.27–0.82)	0.72 (0.43–1.20)	0.39 (0.17–0.90)	1.17 (0.79–1.71)	1.18 (0.68–2.04)
Sensory impairment	1.15 (0.85–1.55)	11.01 (1.45–83.91)	7.76 (0.96–62.82)	- *	- *	- *	1.14 (0.81–1.59)	1.21 (0.82–1.77)
*Eye disease*	0.88 (0.74–1.05)	1.74 (1.06–2.88)	2.69 (1.65–4.38)	4.78 (3.06–7.46)	5.69 (3.66–8.85)	7.36 (4.70–11.55)	0.90 (0.75–1.09)	0.99 (0.78–1.25)
*Hearing impairment*	1.33 (1.11–1.60)	1.21 (0.76–1.91)	1.83 (1.18–2.85)	2.51 (1.68–3.77)	4.01 (2.74–5.88)	5.71 (3.90–8.35)	1.21 (0.99–1.47)	1.11 (0.87–1.42)
Incontinence	0.58 (0.42–0.82)	1.17 (0.48–2.87)	2.49 (1.10–5.63)	3.45 (1.61–7.41)	5.37 (2.56–11.28)	7.86 (3.71–16.65)	0.90 (0.62–1.29)	1.09 (0.72–1.65)
Dizziness	0.73 (0.56–0.96)	1.21 (0.65–2.27)	1.61 (0.87–2.99)	2.16 (1.23–3.80)	3.75 (2.20–6.38)	5.08 (2.95–8.77)	1.15 (0.85–1.57)	1.46 (1.03–2.07)
Dementia	1.11 (0.39–3.11)	-	1.05 (0.06–17.3)	2.46 (0.24–24.8)	5.68 (0.74–43.78)	6.72 (0.83–56.61)	1.37 (0.36–5.28)	3.33 (0.94–11.76)

* Instable result due to too few cases. Ref.: reference; RR: relative risk; CI: confidence intervals; yrs.: years.

**Table 5 ijerph-19-11777-t005:** Risk of multimorbidity.

Variables	Presence of Multimorbidity Wtd. % (95% CI)	Risk of Multimorbidity RR * (95% CI)
Gender		
Women	72.7 (70.1–75.1)	1.00 (Ref.)
Men	73.6 (70.9–76.2)	1.05 (0.98–1.11)
**Age Group**		
60–64 years	57.7 (52.7–62.6)	1.00 (Ref.)
65–69 years	62.4 (57.9–66.8)	1.10 (0.97–1.24)
70–74 years	74.1 (69.5–78.3)	1.27 (1.14–1.43)
75–79 years	78.9 (75.0–82.4)	1.37 (1.24–1.53)
80–84 years	86.9 (83.5–88.6)	1.51 (1.37–1.67)
85+ years	90.8 (86.4–93.9)	1.55 (1.39–1.72)
**SES**		
High	67.1 (63.6–70.5)	1.00 (Ref.)
Middle	73.2 (70.1–76.1)	1.10 (1.03–1.17)
Low	82.2 (77.5–86.3)	1.15 (1.07–1.24)

* Results of Poisson regression, adjusted for gender, age, and socioeconomic status (SES). CI = confidence intervals; RR = relative risk; Wtd. % = weighted %.

**Table 6 ijerph-19-11777-t006:** Care dependency of LAB60+ participants, after weighting, by age group, gender, and socioeconomic status (SES).

Characteristics	Care Dependency		RR ^‡^ (95% CI)
	**No**	**Yes**	
	**(N = 2063)**	**(N = 293)**	
	**Wtd. % (95% CI)**	**Wtd. % (95% CI)**	
**Sex**			
Women	86.3 (84.2–88.1)	13.7 (11.9–15.7)	1.00 (Ref.)
Men	91.3 (89.5–92.8)	8.7 (7.2–10.5)	0.97 (0.95, 0.99) *
**Age Group**			
60–64 years	98.9 (97.4, 99.5)	1.1 (0.5, 2.6)	1.00 (Ref.)
65–69 years	97.8 (96.0, 98.8)	2.2 (1.2, 4.0)	1.01 (1.00, 1.03)
70–74 years	95.2 (92.4, 97.0)	4.8 (3.0, 7.6)	1.03 (1.01, 1.06) *
75–79 years	89.9 (86.9, 92.3)	10.1 (7.7, 13.1)	1.08 (1.05, 1.11) *
80–84 years	77.3 (72.7, 81.3)	22.7 (18.7, 27.3)	1.18 (1.13, 1.22) *
85+ years	53.8 (47.3, 60.3)	46.2 (39.7, 52.7)	1.38 (1.32, 1.46) *
**Socioeconomic Status**			
High	95.1 (93.3, 96.4)	4.9 (3.6, 6.7)	1.00 (Ref.)
Middle	91.3 (89.3, 93.1)	8.7 (7.0, 10.7)	1.04 (1.02, 1.06) *
Low	75.5 (70.3, 80.1)	24.5 (19.9, 29.7)	1.12 (1.08, 1.16) *

Wtd.% = weighted %; n/a = not applicable; RR = relative risk; CI = confidence intervals.* Statistically significant at *p* < 0.05; ^‡^ Poisson regression.

**Table 7 ijerph-19-11777-t007:** Health-related quality of life, well-being, and depressivity of survey respondents, after weighting, by gender.

Characteristics	All	Women	Men
	(N = 2399)	(N = 1254)	(N = 1136)
	Wtd. % (95% CI)	Wtd. % (95% CI)	Wtd. % (95% CI)
**SF-8**			
GH Mean (95% CI)	44.5 (44.3, 44.8)	44.2 (43.8, 44.6)	44.9 (44.6, 45.3)
PF Mean (95% CI)	43.8 (43.5, 44.2)	43.3 (42.8, 43.9)	44.4 (43.9, 44.9)
RP Mean (95% CI)	44.7 (44.3, 45.0)	44.1 (43.6, 44.6)	45.2 (44.7, 45.8)
BP Mean (95% CI)	48.3 (47.9, 48.7)	47.3 (46.8, 47.9)	49.4 (48.9, 50.0)
VT Mean (95% CI)	47.6 (47.3, 47.9)	46.9 (46.5, 47.4)	48.3 (47.9, 48.7)
SF Mean (95% CI)	46.1 (45.7, 46.5)	45.2 (44.7, 45.8)	47.0 (46.4, 47.6)
MH Mean (95% CI)	48.0 (47.6, 48.4)	46.7 (46.2, 47.3)	49.4 (48.9, 49.9)
RE Mean (95% CI)	45.7 (45.4, 46.0)	45.0 (44.5, 45.5)	46.5 (46.0, 46.9)
PCS-8 Mean (95% CI)	44.5 (44.0, 44.9)	43.9 (43.3, 44.5)	45.1 (44.5, 45.6)
MCS-8 Mean (95% CI)	48.2 (47.7, 48.6)	46.8 (46.2, 47.4)	49.6 (49.1, 50.2)
**WHO-5**			
Mean (95% CI)	58.3 (57.3, 59.2)	55.3 (54.0, 56.7)	61.4 (60.1, 62.7)
Depressivity			
No	66.3 (64.4, 68.2)	61.9 (59.0, 64.6)	71.1 (68.3, 73.8)
Yes	33.7 (31.8, 35.6)	38.1 (35.4, 41.0)	28.9 (26.2, 31.7)

Wtd.% = weighted %; CI = confidence interval; n/a = not applicable. Cell percentages are based on weighted data for everyone in that age group. N at top of each column is the unweighted number of respondents in that group. **GH**—general health; **PF**—physical functioning; **RP**—role-physical; **BP**—bodily pain; **VT**—vitality; **SF**—social functioning; **RE**—role-emotional; **MH**—mental health; **PCS**—physical summary scale; **MCS**—mental summary scale.

**Table 8 ijerph-19-11777-t008:** Associations between gender, age, and socioeconomic status on health-related quality of life and well-being.

Variables	PCS-8 β ^†^ (95% CI)	MCS-8 β ^†^ (95% CI)	Well-Being β ^†^ (95% CI)	Depressivity RR ^‡^ (95% CI)
**Sex**				
Women	1.00 (Ref.)	1.00 (Ref.)	1.00 (Ref.)	1.00 (Ref.)
Men	0.36 (−0.51, 1.22)	2.43 (1.50, 3.36) *	5.16 (3.06–7.26) *	0.78 (0.68–0.89) *
**Age Group**				
60–64 years	1.00 (Ref.)	1.00 (Ref.)	1.00 (Ref.)	1.00 (Ref.)
65–69 years	0.18 (−1.11, 1.47)	1.82 (0.44, 3.20) *	4.14 (1.00, 7.28) *	0.78 (0.62–0.98) *
70–74 years	−2.74 (−4.18, −1.30)*	1.82 (0.29, 3.36) *	3.63 (0.13, 7.14) *	0.89 (0.70–1.13)
75–79 years	−4.84 (−6.16, −3.52) *	0.15 (−1.26, 1.57)	−1.06 (−4.26, 2.15)	1.12 (0.91–1.39)
80–84 years	−7.29 (−8.77, −5.82) *	−0.13 (−1.71, 1.44)	−3.74 (−7.31, −0.17) *	1.14 (0.92–1.42)
85+ years	−9.98 (−11.7, −8.25) *	0.05 (−1.79, 1.90)	−5.72 (−9.86, −1.59) *	1.28 (1.01–1.63) *
**SES**				
High	1.00 (Ref.)	1.00 (Ref.)	1.00 (Ref.)	1.00 (Ref.)
Middle	−2.59 (−3.53, −1.64) *	−2.00 (−3.00, −0.99) *	−4.40 (−6.67, −2.12) *	1.24 (1.06–1.45) *
Low	−5.11 (−6.44, −3.78) *	−3.17 (−4.59, −1.75) *	−7.60 (−10,79, −4.41) *	1.43 (1.18–1.73) *

* Statistically significant (*p* < 0.05); all effect estimates adjusted for gender, age group, and socioeconomic status (SES). ^†^ Multivariate linear regression. ^‡^ Poisson regression. RR = relative risk; β = regression coefficient; Ref. = reference; PCS-8: physical component score; MCS-8: mental component score; WHO-5: 5-item World Health Organization Well-Being Index.

## Data Availability

The data presented in this study are available on request from the corresponding author. The data are not publicly available due to privacy reasons.

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
