# Peer review of "Health and Care Dependency of Older Adults in Dresden, Germany: Results from the LAB60+ Study"

_ijerph, 2022, doi:10.3390/ijerph191811777_

Round 1

Reviewer 1 Report

The subject of study of this paper is very timely given the aging of the population in today's societies. This article had also been well organized. Some commends would be provide for the authors as follows:

Please add more details to explain why this study chose this city, Dresden. (Line 57-62)

The authors mentioned the interview on Line 90-92. What is the main purpose to design this interview? What kind of results were obtained from the interviews?

The treatment and analysis of quantitative data is very detailed. Only the format of Table 8 is not consistent to other tables.

Discussion section compared the results of this study to the German statistical reports. Although major results were in accordance, the authors may provide the possible reasons or special characteristics in Dresden for the difference results. 

Conclusion section is a little bit too short and general. The research performed with a particular population and location. I would suggest highlighting the main research contributions of this study and also provide the suggestions for the future studies. 

Reviewer 2 Report

It has been a pleasure to review the manuscript “Health and care dependency of older adults in Dresden, Germany: results from the LAB60+ Study” submitted to the International Journal of Environmental Research and Public Health.

The manuscript consists in a public health research work that addresses the relationship between health and health care and age, gender and socio-economic status in a sample of individuals aged 60 and above in the city of Dresden. The main contribution of the study is the specific focus in a local environment and the use of an ad hoc survey.

From my point of view, the contribution of the work to current knowledge is not very large. Nevertheless, the use of new data gives the study some added value. It is also worth mentioning that the authors perform quantitative analysis in a quite competent way, in line with the technical standards of the profession. Another strength of the article is that it compares its results to the ones in other works or the information available for the whole country.

On the basis of my comments above, my overall assessment of the manuscript is positive. Nevertheless, I have some minor comments that can help to improve the paper:

(1) It could be a good idea to elaborate more on the reasons why a city-centred study is valuable and increase our understanding of the issues at stake (lines 57-62).

(2) The aim of the study is descriptive. Whereas it is clear in some places of the article that the authors explore correlations or associations (and not causal effects), in other places, the wording suggests that they are looking at causal relationships (which is not possible in this non-experimental framework in the case of socio-economic status, as there is a two-way causality relationship between income and health). I would suggest avoiding references to a possible causal interpretation of the results, making clear that the paper explores correlations in all cases.

(3) The article is overall well written. Nevertheless, I think that the writing would improve if the authors would reduce the use of the passive voice.

(4) There is some recent Social Sciences literature that explores the additional burden that people with health and disability problems means for individuals in terms of deprivation (i.e., for a certain SES level, material well-being is lower for this individuals). I think that the introduction would improve if it included a very brief paragraph or a few sentences (for instance, in the second paragraph of the introduction), commenting on it. For this purpose, I would recommend citing, for instance, Antón et al. (2016) focused on developed European countries, Suhrcke (2019) and Morgon et al. (2017), centred on middle- and low-income countries.

References cited in the review

—Antón, J.-I., Braña, F. J., & Muñoz de Bustillo, R. (2016). An analysis of the cost of disability across Europe using the standard of living approach. SERIEs—Journal of the Spanish Economic Association, 7(3), 281–306 (2016). https://doi.org/10.1007/s13209-016-0146-5

— Banks, L. M., Kuper, H., & Polack, S. (2017). Poverty and disability in low- and middle-income countries: A systematic review. PloS ONE, 12(12). e0189996. https://doi.org/10.1371/journal.pone.0189996

— Surhcke, M. (2019). Disability and Economic Development. In A. Jones (Ed.), Oxford Research Encyclopedia of Economics and Finance. Oxford University Press. https://doi.org/10.1093/acrefore/9780190625979.013.39

Round 2

Reviewer 1 Report

The authors did revised the manuscript very well, provided more details about the study and improved the quality of this paper.